# Levels and functionality of Pacific Islanders' hybrid humoral immune response to BNT162b2 vaccination and delta/omicron infection: A cohort study in New Caledonia

Catherine Inizan[1]*, Adrien Courtot[2], Chloé Sturmach[3,4], Anne-Fleur Griffon[1], Antoine Biron[5], Timothée Bruel[6,7,8], Vincent Enouf[3,4], Thibaut Demaneuf[9¤a], Sandie Munier[4], Olivier Schwartz[7,8], Ann-Claire Gourinat[5¤b], Georges Médevielle[2], Marc Jouan[1], Sylvie van der Werf[3,4], Yoann Madec[10], Valérie Albert-Dunais[11], Myrielle Dupont-Rouzeyrol[1]

1 Dengue and Arboviroses – Research and Expertise Unit - Institut Pasteur in New Caledonia - Pasteur Network, Dumbéa-sur-Mer, New Caledonia, 2 Provincial Office for Health and Social Action of the South Province (Direction Provinciale de l'Action Sanitaire et Sociale en Province Sud), Nouméa, New Caledonia, 3 National Reference Center for Respiratory Viruses, Institut Pasteur, Université Paris Cité, Paris, France, 4 Molecular Genetics of RNA Viruses Unit, Institut Pasteur, Université Paris Cité, CNRS UMR3569, Paris, France, 5 New Caledonia Territorial Hospital, Dumbéa-sur-Mer, New Caledonia, 6 Antiviral Activities of Antibodies Group, Institut Pasteur, Université Paris Cité, CNRS UMR3569, Paris, France, 7 Virus and Immunity Unit, Institut Pasteur, Université Paris Cité, CNRS UMR3569, Paris, France, 8 Vaccine Research Institute, Créteil, France, 9 Social and Sanitary Agency of New Caledonia (Agence Sanitaire et Sociale de Nouvelle-Calédonie), Nouméa, New Caledonia, 10 Epidemiology of Emerging Diseases, Institut Pasteur, Université de Paris, Paris, France, 11 New Caledonia Specialized Hospital, Nouméa, New Caledonia

¤a Current address: Pacific Community SPC – Public Health Division – B.P. D5, Nouméa, New Caledonia
¤b Current address: New Caledonia Health Authorities (Direction des Affaires Sanitaires et Sociales de Nouvelle-Calédonie), Nouméa, New Caledonia
* cinizan@pasteur.nc

**Data Availability Statement:** All relevant data are within the manuscript and its Supporting information files. Requests for reagents and resources should be directed to and will be fulfilled by the department in charge of clinical

## Abstract

### Background

Pacific Islanders are underrepresented in vaccine efficacy trials. Few studies describe their immune response to COVID-19 vaccination. Yet, this characterization is crucial to re-enforce vaccination strategies adapted to Pacific Islanders singularities.

### Methods and findings

We evaluated the humoral immune response of 585 adults, self-declaring as Melanesians, Europeans, Polynesians, or belonging to other communities, to the Pfizer BNT162b2 vaccine. Anti-spike and anti-nucleoprotein IgG levels, and their capacity to neutralize SARS-CoV-2 variants and to mediate antibody-dependent cellular cytotoxicity (ADCC) were assessed across communities at 1 and 3 months post-second dose or 1 and 6 months post-third dose. All sera tested contained anti-spike antibodies and 61.3% contained anti-nucleoprotein antibodies, evidencing mostly a hybrid immunity resulting from vaccination and SARS-CoV-2 infection. At 1-month post-immunization, the 4 ethnic communities exhibited no significant differences in their anti-spike IgG levels ($p$ value = 0.17, in an

investigations at the Institut Pasteur in New Caledonia, (ipnc-etudeclinique@pasteur.fr).

**Funding:** This study received financial support from the ANRS | Emerging infectious diseases (Programme CAPNET, COVCAL project n° 103487514 awarded to MDR), the Institut Pasteur (Calmette and Yersin Grant awarded to CI), the Pasteur Network (URGENCE COVID-19 funding awarded to MDR), the Institut Pasteur in New Caledonia (funding awarded to CI), the Specialized Hospital from New Caledonia (funding awarded to VAD), the Provincial Office for Health and Social Action in the South Province in New Caledonia (DPASS-Sud, funding awarded to GM) and the Regional Hospital Federation in the South Pacific (FHF, funding awarded to VAD). The funders had no role in study design, data collection and analysis, decision to publish or preparation of the manuscript.

**Competing interests:** The authors have declared that no competing interests exist.

**Abbreviations:** ADCC, antibody-dependent cellular cytotoxicity; AU, Arbitrary Units; BMI, body mass index; DMEM, Dulbecco's Modified Eagle Medium; IQR, interquartile range; PBS, phosphate buffer saline; ROC, receiver operating characteristic.

univariate linear regression model), in their capacity to mediate omicron neutralization ($p$ value = 0.59 and 0.60, in an univariate logistic regression model at 1-month after the second and third dose, respectively) and in their capacity to mediate ADCC ($p$ value = 0.069 in a multivariate linear regression model), regardless of the infection status. Anti-spike IgG levels and functionalities of the hybrid humoral immune response remained equivalent across the 4 ethnic communities during follow-up and at 6 months post-third dose.

## Conclusions

Our study evidenced Pacific Islander's robust humoral immune response to Pfizer BNT162b2 vaccine, which is pivotal to re-enforce vaccination deployment in a population at risk for severe COVID-19.

## Trial registration

This trial has been register in ClinicalTrials.gov (ID: NCT05135585).

---

## Author summary

### Why was this study done?

- Pacific Islanders are at risk for severe COVID-19, in part owing to lifestyle and high rate of comorbidities.

- In a few studies, ethnicity had no impact on antibody levels following Pfizer BNT162b2 vaccine in Pacific Islanders; in these studies, however, follow-up was reduced and no distinction was made between the different Pacific ethnic communities.

- Characterizing the levels and functionality of Pacific Islanders humoral immune response to Pfizer BNT162b2 vaccine is pivotal to adapt vaccination regimens to regional singularities.

### What did the researchers do and find?

- The current study enrolled 585 participants from New Caledonia self-declaring as Melanesians, Polynesians, Europeans, or belonging to other communities. Antibody levels and their functionality were characterized at the following time points: (1) 1 and/or 3 months after a second dose of the Pfizer BNT162b2 vaccine was administered; or (2) 1 and/or 6 months after the third dose of the vaccine.

- Approximately 61.3% of all sera contained traces of previous SARS-CoV-2 infection; the cohort's immune response was thus mostly hybrid, resulting from both vaccination and infection.

- At all time points analyzed, the hybrid humoral immune response remained equivalent across the Melanesian, Polynesian, European, and other communities: anti-spike antibody levels, which mirror the strength of the immune response, but also the capacity to mediate omicron neutralization and the capacity to mediate antibody-dependent

cellular cytotoxicity (ADCC) were not significantly different across communities (*p* values >0.05).

## What do these findings mean?

- The lack of community effect, despite different prevalence of obesity, suggests that SARS-CoV-2 hybrid humoral response does not differ between communities.

- Such robust and prototypical humoral immune response to Pfizer BNT162b2 vaccine and SARS-CoV-2 in Pacific Islanders is of prime importance to foster booster doses roll-out, with the aim to better protect this understudied population which is vulnerable to severe COVID-19.

- Biological confirmation of ethnicity and a larger sample size in the Polynesian community would enhance the resolution of our findings.

## Introduction

Covering 8.5 million km$^2$ in a maritime area of 80 million km$^2$, emerged Pacific territories excluding Australia are populated by ≈17 million inhabitants, among which 11.6 million belong to communities of Non-European Non-Asian descent [1]. Pacific Islanders are at risk of severe respiratory infections, including influenza [2] and COVID-19 [3]. As vaccination protects from severe COVID-19 in cohorts mostly composed of people of European descent [4], it is pivotal to assess whether Pacific Islanders' humoral response to anti-COVID-19 vaccination is equivalent to the one of people of European descent.

New Caledonia is a French overseas territory of ≈270,000 inhabitants in the South Pacific region, where 41.2% self-declare as Melanesian, 24% as European, 8.3% as Polynesian, and 26.5% as belonging to other communities [5].

A zero COVID policy protected New Caledonia from early SARS-CoV-2 spread. In September 2021, the first epidemic peak was caused by the delta variant. Two larger epidemic peaks due to omicron BA.1 and BA.4/5 occurred in 2022 (S1 Fig). Almost 80,000 infections and 314 deaths were detected [6,7]. Vaccination roll-out, involving almost exclusively Pfizer BNT162b2 vaccine, was initiated in January 2021. As of May 2023, 70% of the entire population had received at least 1 dose, 68% had received at least 2 doses, and 38% had received a booster dose [8].

Considering ethnic representation in COVID-19 vaccine trials is crucial [9] to adapt vaccination regimens to regional singularities. However, in Pfizer BNT162b2 efficacy trial, the cohort included only 0.2% Native Hawaiians/other Pacific Islanders [10], precluding to assess vaccine efficacy in this population. As a consequence, data on anti-SARS-CoV-2 immunity in vaccinated Pacific populations remain scarce. A search on Pubmed on October 24, 2023 with the following query: ((((ethnicity) OR (race)) AND ((Pacific) OR (Pacific Islanders) OR (Oceania)) AND ((immune response) OR (antibody))) AND (Vaccine)) AND ((COVID-19) OR (SARS-CoV-2)) returned 17 results, of which only 3 described the immune response of New Zealand adults, including 80 or 81 Pacific Islanders, and Australian First Nation people to COVID-19 vaccination. In a New Zealand cohort study, the 80 Pacific people included had lower anti-SARS-COV-2 antibody titers than Europeans and Maoris 28 days after 2 doses of BNT162b2 vaccine. However, when adjusting for the higher prevalence of diabetes in Pacific

people, antibody titers and variant neutralization did not differ by ethnicity [11]. Similarly, humoral and cellular responses to Pfizer BNT162b2 vaccine were similar in Australian First Nations and Non-Indigenous people [12]. However, the humoral immune response of Melanesian and Polynesian people to COVID-19 vaccination was never formally assessed, especially in large cohorts. Characterization of Pacific Islanders' immune response to COVID-19 vaccination is nonetheless of paramount importance to adapt vaccination regimens, including booster doses roll-out, to better protect Pacific Islanders.

Anti-spike antibody levels, neutralization capacity and antibodies functionality, including antibody-dependent cellular cytotoxicity (ADCC) [13], play an important role in COVID-19 pathogenesis. Neutralization correlates to vaccine efficacy [14]. ADCC was higher in hospitalized or severe patients who survived than in fatal cases [13,15], suggesting its protective role.

Adaptation of regional vaccination strategies to Pacific populations' singularities requires to evaluate COVID-19 vaccines' immunogenicity and hybrid immunity upon breakthrough infection in Pacific populations. The current study postulated that the hybrid humoral response to Pfizer BNT162b2 vaccine was equivalent across the 4 ethnic communities in New Caledonia. We aimed to characterize Pacific Islanders' hybrid humoral response to BNT162b2 vaccine and SARS-CoV-2 infection, by evaluating anti-nucleoprotein IgG levels, anti-spike (anti-S) IgG levels, the neutralization capacity, and the ability to mediate ADCC of sera longitudinally collected in a cohort of 585 vaccinated adults from New Caledonia self-declaring as Melanesians, Polynesians, Europeans, or belonging to other communities.

## Methods

### Ethics

The current study was conducted in compliance with the Declaration of Helsinki principles. The study was recorded on clinicaltrials.gov (ID: NCT05135585). Samples came from participants who gave their written informed consent as part of this study. A prospective study protocol was set up ahead of study implementation. This study protocol was approved by the "Comité de Protection des Personnes Ile-de-France III" (n° ID-RCB 2021-A01949-32, CPP n° 4003-I, November 9, 2021, Am8898-1-4003 January 17, 2022, Am9503-2-4003 April 4, 2022, and Am9508-3-4003 June 21, 2022) and by the Consultative Ethics Committee of New Caledonia.

### Cohort and study design

The current study follows the STROBE (**STrengthening the Reporting of OBservational studies in Epidemiology)** reporting guidelines (https://www.strobe-statement.org/). The filled STROBE checklist is shown (S1 STROBE Checklist). Enrolment of adults ($\geq$18 years) vaccinated with 2 or 3 doses of the Pfizer BNT162b2 vaccine took place in 3 phases in immunization and health centers in the South Province of New Caledonia (S1 Fig). Between December 2021 and January 2022, adults who received the second dose of the Pfizer BNT162b2 vaccine were enrolled (Fig 1A). Between January 2022 and March 2022, when information campaigns largely promoted immunization with a third dose, adults who received a third dose were enrolled (participants enrolled at the third dose, Fig 1B). Enrolment in immunization centers took place every working day to maximize the number of enrolled participants at the peak of vaccination roll-out. Between March and August 2022, when the number of third doses delivered decreased, individuals having received a third dose within the last 6 months were enrolled through targeted information campaigns (participants enrolled remotely from the third dose, Fig 1B). Sample size calculation for this equivalence study was based on a power of 85% to detect changes of more than 5% in the fraction of participants developing anti-S IgG in

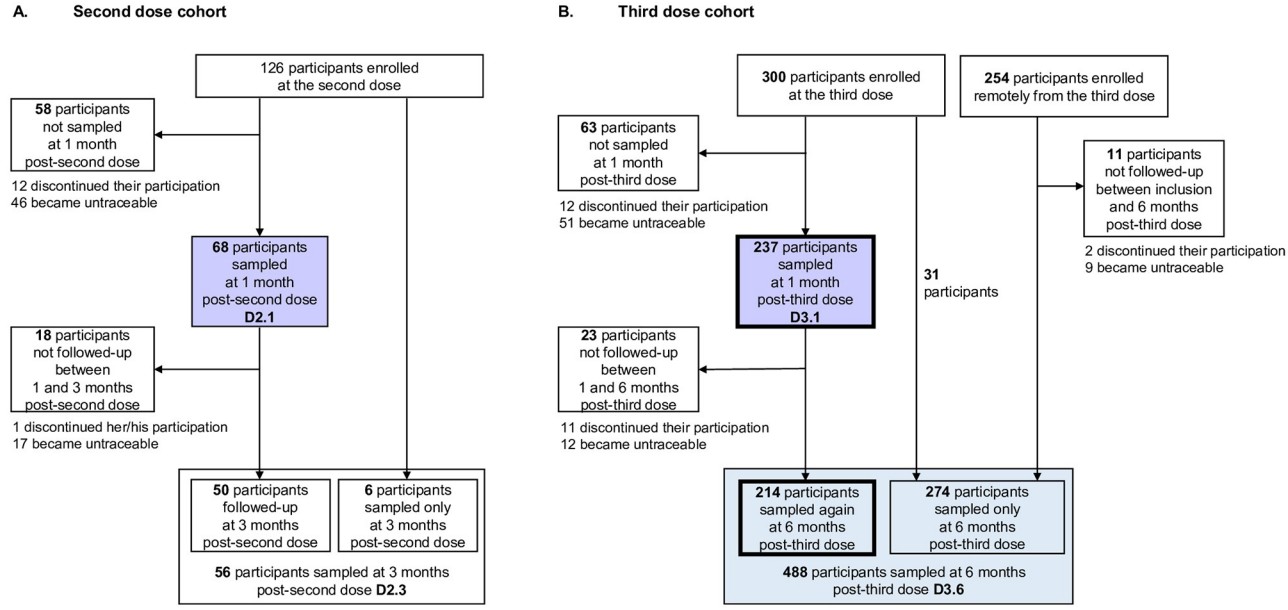

**Fig 1. STROBE (STrengthening the Reporting of OBservational studies in Epidemiology) diagrams of the second- and third-dose cohorts.** (A) The second dose cohort is composed of 126 enrolled participants, among which 68 were sampled 1 month after their second dose (purple square). (B) The third-dose cohort is composed of 554 enrolled participants, among which 237 were sampled 1 month after their third dose (purple square), 214 were followed-up at 1 and 6 months after their third dose (bold frames) and 488 were sampled at 6 months after their third dose (blue square).

response to Pfizer BNT162b2 vaccination, with an alpha risk of 5%, resulting in the target to enroll 141 participants in each of the Melanesian, European, Polynesian, and "Other" community. Based on a weekly monitoring of the number of participants enrolled in each of the 4 communities, enrolment was targeted in immunization and health centers located in specific geographical areas mostly populated by the communities of interest (ISEE, 2019 census). Individuals fulfilling inclusion/exclusion criteria were presented the research project by a member of the investigation team; they were given time to read the information notice and decide whether they wanted to participate in the study. During the inclusion visit, inclusion criteria were verified: having received the second dose between 3 and 12 weeks after the first dose, and the third dose between 3 and 6 months after the second dose were inclusion criteria. Documented SARS-CoV-2 infection at any time before inclusion (serologic, PCR or antigenic testing) was an exclusion criterion. Pregnant or breast-feeding women, immunosuppressed individuals, individuals unable to answer the questionnaire, and those under protective measures were excluded. Following signature of the consent form, clinico-epidemiological data (age, self-declared gender, self-declared ethnicity, medical history, height, and weight) were recorded in REDCap (Vanderbilt University, United States of America). Personal contact details were collected to organize the follow-up visits.

For all participants, follow-up visits were to be performed 1 month (+/− 10 days) and 3 months after the second dose or 1 month (+/− 10 days) and/or 6 months after the third dose. The week preceding the planed sampling period, the investigation team arranged a home visit by a qualified nurse by contacting all participants by phone, adapting as best as possible to participants' availability. Upon these visits, a 4-ml venous blood sample was collected. Sera were separated from blood samples by centrifugation and stored at −20°C until subsequent analyses.

Sampling took place between January and August 2022. Not all participants enrolled in the study participated to the follow-up: some participants became untraceable (impossibility to

contact the participants, absence of presentation at follow-up visits), others refused to pursue their participation in the study (expressed during phone contacts), as shown in Fig 1.

## Cells and reagents

HEK 293T-hACE2 cells were cultured at 37°C under 5% $CO_2$ in complete culture medium composed of Dulbecco's Modified Eagle Medium (DMEM, Gibco) complemented with 10% fetal calf serum (Gibco), 1% penicillin/streptomycin (Gibco), and 2 mM L-Glutamine (Gibco). Cells were detached and resuspended using 0.05% Trypsin-EDTA (Gibco).

Recombinant SARS-CoV-2 Wuhan spike and nucleocapsid proteins were kindly provided by S. Petres [16]. Secondary lama antibodies VHH anti-human IgG Fc coupled to NanoKAZ were kindly provided by Thierry Rose. Z108 NanoKAZ substrate was kindly provided by Yves Janin.

## Pseudotyped viruses

Pseudotyped lentiviral vectors enveloped with the spike protein from Wuhan, delta, omicron BA.1, or omicron BA.4/5 SARS-CoV-2 variants were produced as previously described [17]. Briefly, HEK 293T cells were co-transfected with a packaging plasmid Gag-Pol (NR-52517, BEI Resources), Helper Plasmids Tat1b (NR-52518, BEI Resources), and Rev1b (NR-52519, BEI Resources), a transfer plasmid carrying the gene encoding the spike glycoprotein of Wuhan-Hu-1 (GenBank: NC_045512, plasmid NR-52514, BEI Resources), delta, omicron BA.1 or omicron BA.4/5 SARS-CoV-2 variants and a Lentiviral Backbone carrying a luciferase reporter gene Luc2 and a ZsGreen gene (pHAGE-CMV-Luc2-IRES-ZsGreen-W, NR-52516, BEI Resources). At 48 h post-transfection, supernatants were collected, clarified by centrifugation, and frozen at −80°C.

## LuLISA experiments

Anti-spike and nucleoprotein IgG levels were assessed in LuLISA experiments performed using recombinant SARS-CoV-2 spike and nucleoprotein proteins from the ancestral Wuhan strain. Sera were decomplemented at 56°C for 35 min. White 384-well tissue-culture plates (Greiner Bio-one) were coated with recombinant SARS-CoV-2 Wuhan spike or nucleocapsid proteins at 1 µg/ml in phosphate buffer saline (PBS, Gibco) through 4 h incubation at room temperature. Plates were washed with PBS containing 0.1% Tween (Tween20, Sigma-Aldrich) using a plate washer Zoom microplate Washer (Titertek Berthold). Sera were diluted 1/200 in PBS containing 1% skimmed milk and 0.1% Tween using a Tecan Fluent 780 and incubated in duplicate for 2 h in spike- or nucleoprotein-coated plates. Plates were washed with PBS containing 0.1% Tween. Secondary VHH lama antibodies anti-human IgG Fc coupled to NanoKAZ were diluted at 20 ng/ml in PBS containing 1% skimmed milk and 0.1% Tween and added to the plates. After 1 h 30 incubation, Z108 NanoKAZ substrate was added to the plates at 6 µg/ml. Bioluminescence was measured using Mithras$^2$ LB943 Multimode reader luminometer (Berthold Technologies). Presence of anti-S IgG above the 0.25 AU was indicative of an immunization against SARS-CoV-2. Past COVID-19 infection was determined by the presence of anti-nucleocapsid (anti-N) IgG above 0.3 AU. Furthermore, for participants who underwent a longitudinal sampling, breakthrough infections were defined as an anti-N IgG negative sample at the first sampling time point and an anti-N IgG positive sample at the second sampling time point; reinfections were defined based on an increase in the anti-N IgG levels between the first and second sampling time points.

## Pseudovirus neutralization assay

Capacity of participants' sera to neutralize either SARS-CoV-2 Wuhan, delta, omicron BA.1 or omicron BA.4/5 variants was assessed in 4 pseudovirus neutralization assays, each using a different pseudotyped virus enveloped with the spike protein of one of the following variants: ancestral Wuhan strain; delta variant, omicron BA.1 variant, or omicron BA.4/5 variant. Decomplemented sera (56˚C for 35 min) were diluted 1/40 and co-incubated with pseudotyped lentiviral vectors at room temperature for 50 min in complete culture medium prepared using DMEM without phenol red (Gibco). The mixture was then plated in white 96-well tissue-culture plates (Greiner Bio-One). A suspension of 20,000 HEK 293T-hACE2 cells was added in each well. Single-cell suspensions were prepared in Dulbecco's PBS (DPBS) containing 0.1% EDTA (Promega) to preserve hACE2 protein integrity. Serum-lentivirus mixes and addition of the cells were all performed using a Tecan Fluent 780. After 72 h, Bright-Glo Luciferase substrate (Promega) was added to the cells (1:1). Bioluminescence was measured using Centro XS3 LB 960 or Mithras$^2$ LB943 Multimode reader luminometers (Berthold Technologies).

A set of negative sera was used to determine the threshold as the average luminescence level of these negative sera minus 3 standard deviations.

The percentage of neutralization was expressed as $(1 - (\text{measured value/threshold})) \times 100$. A participant's serum was considered able to neutralize omicron variant if a 1/40 dilution of his/her serum exhibited a $\geq$90% neutralization of either omicron BA.1 and/or omicron BA.4/5 pseudoviruses in pseudovirus neutralization assays (i.e., neutralization titer >40).

## Antibody-dependent cellular cytotoxicity (ADCC)

CD16 activation was quantified as a surrogate assay to measure sera's capacity to mediate ADCC. Capacity of participants' sera to activate CD16 was quantified using 293T cells stably expressing the spike protein from SARS-CoV-2 ancestral strain Wuhan. ADCC was quantified using the ADCC Reporter Bioassay (Promega). Briefly, spike-expressing cells ($3 \times 10^4$ per well) were co-cultured with Jurkat-CD16-NFAT-rLuc cells ($3 \times 10^4$ per well) in presence or absence of sera (diluted 1/30). Luciferase was measured after 18 h of incubation using an EnSpire plate reader (PerkinElmer). ADCC was measured as the fold induction of Luciferase activity compared to the "no serum" condition. For each serum, the control condition (cells transfected with an empty plasmid) was subtracted to account for inter-individual variations of the background. We previously reported correlations between the ADCC Reporter Bioassay titers and an ADCC assay based on primary NK cells and cells infected with an authentic virus [13].

## Statistical analyses

Quantitative variables were characterized by the median, interquartile range (IQR), and range. Age was categorized into 3 groups: 18–39, 40–64, and $\geq$65 years. Participants with a body mass index (BMI) strictly below 18.5 kg/m$^2$, comprised between 18.5 and 24.9 kg/m$^2$, comprised between 25 and 29.9 kg/m$^2$ or over 30 kg/m$^2$ were considered as underweight, normal weight, overweighted, or obese, respectively.

Proportions were compared using Chi$^2$ test. Continuous variables were compared using Kruskal–Wallis test. A ROC (receiver operating characteristic) curve analysis was used to identify the anti-S IgG threshold that best discriminated participants able to neutralize omicron with an optimal specificity.

The effect of immunization dose, infection status, self-declared gender, age, BMI category, presence of comorbidities and self-declared ethnicity on the levels of anti-S IgG, and CD16 activation was investigated using linear regression models. The effect of self-declared ethnicity

on anti-N IgG levels was investigated using linear regression models. Their effect on omicron neutralization capacity was investigated using logistic regression models. Linear regression analyses and logistic regression analyses were conducted at 1-month and 6-month post-immunization, and also on the difference observed between these 2 time points. A multivariate model including all variables was built. Subsequently, a backward stepwise procedure was used to identify factors that remained independently associated with the outcome. A $p$ value $<0.05$ was considered significant. Only participants sampled at the time points of interest were considered in the corresponding analysis. Sensitivity analyses not considering participants self-declaring as belonging to other communities were performed at 1-month post-immunization. All statistical analyses were performed using R software (version R 4.2.1 GUI 1.79 High Sierra build (8095)).

## Results

### Hybrid humoral response was highly prevalent in the cohort

This study considered 585 participants that provided blood samples, among which 148 self-declared as Melanesians, 189 as Europeans, 82 as Polynesians, and 166 as belonging to other communities. The study period was short, from December 2021 to August 2022, thus limiting the risk of selection bias between communities. As the targeted sample size for each community was not reached in either of the second- and third-dose cohorts, participants from all 4 communities were enrolled along the entire enrolment period (S2 Fig). The proportion of women was 55.9% (327/585). Median age was 51 years (range 19 to 96) and median BMI was 27.5 (IQR [23.4, 32.0]). The proportion of participants with comorbidities was 49.9% (292/585). Median duration between the first and second dose was 21 days (IQR [21, 22]) and median duration between the second and the third dose was 147 days (IQR [128; 183]).

Sixty-eight participants were sampled at 1 month post-second dose (D2.1) and 56 participants were sampled at 3 months post-second dose (D2.3), of whom 50 participants were sampled at both time points (Fig 1). A total of 237 participants were sampled at 1 month post-third dose (D3.1) and 488 participants were sampled at 6 months post-third dose (D3.6), of whom 214 participants were sampled at both time points (Fig 1). Overall, 38 (5.6%) participants refused to pursue their participation in the study and 135 (19.9%) participants were untraceable (Fig 1). There was no missing clinico-epidemiological data. Overall, 849 blood samples were collected and analyzed. Anti-S IgG levels and omicron neutralization could be analyzed in all samples. ADCC analyses were conducted on all, but 2 samples collected at 1-month post-immunization and on all samples collected at 6 months after the third dose from participants for which a sample was also collected at 1-month post-immunization.

Importantly, anti-S IgG were detected in all samples at all time points. Therefore, the planned analysis of the fraction of participants presenting anti-S IgG was no longer relevant. We investigated the levels of anti-S IgG, the neutralization capacity, and the capacity to mediate ADCC.

Eighty-five participants (14.5%) reported a positive COVID-19 test at one of the sampling time points. Participants exhibiting anti-N IgGs, at D2.1, D2.3, D3.1, and D3.6, were 47 (69.1%), 43 (76.8%), 150 (63.3%), and 280 (57.4%), respectively (Fig 2). Detection of anti-N IgG evidenced a previous or current SARS-CoV-2 infection, whether it occurred prior to immunization or between immunization and sampling. In the 41 participants from the second dose cohort, with anti-N antibodies at any time point and followed up between D2.1/D2.3, 23 infections prior to the first sampling, 9 re-infections, and 9 new infections despite immunization (breakthrough infections) were identified. In the 172 participants from the third dose cohort, with anti-N antibodies at any time point and followed up between D3.1/D3.6, 59

**Distribution of the 849 samples**

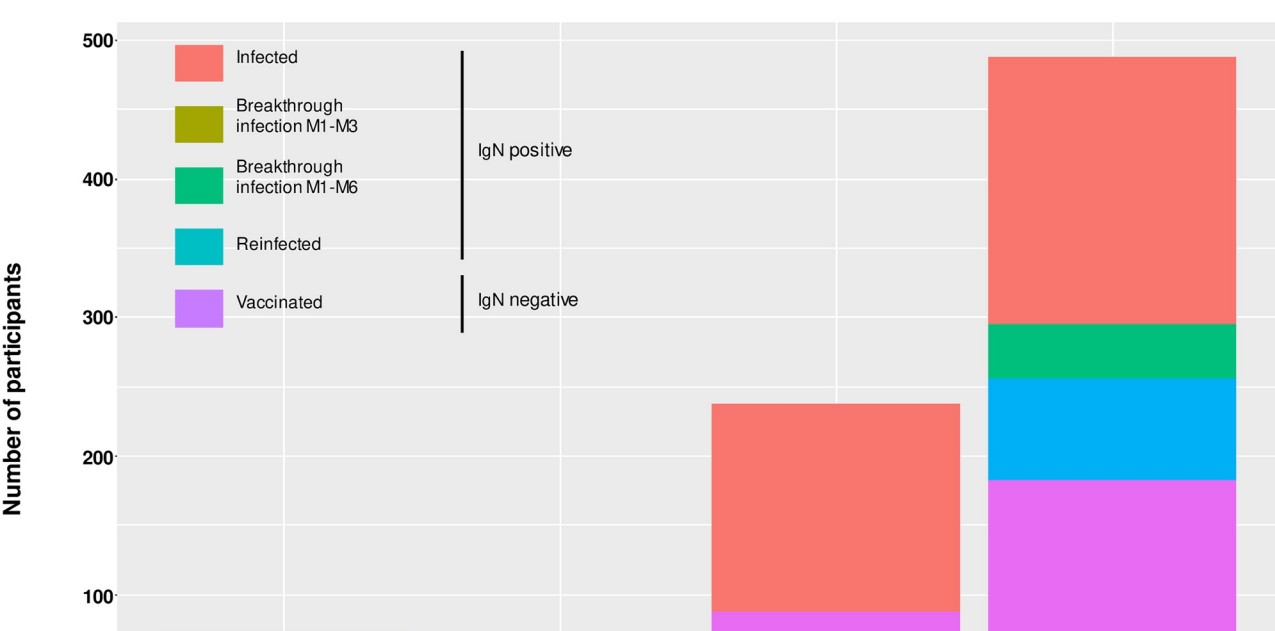

**Fig 2. Description of the sample set.** The 849 samples are divided into anti-N IgG (IgN) negative samples coming from vaccinated participants with no traces of previous infection (violet), and anti-N IgG positive samples coming from infected participants (red), breakthrough infections between 1 and 3 months post-second dose (brown), breakthrough infections between 1 and 6 months post-third dose (green) or reinfected participants (blue).

infections prior to the first sampling, 73 re-infections, and 40 breakthrough infections were identified. Subsequent multivariate regression and logistic models were adjusted for infection status.

## Humoral response at 1-month postvaccination is equivalent across ethnic communities

Among the 305 participants sampled at 1-month post-immunization (68 and 237 sampled at D2.1 and D3.1, respectively), ethnicity was distributed as follows: 55 Melanesians, 79 Europeans, 55 Polynesians, and 116 participants belonging to other communities (S1 Table). The proportion of women was 54.6%. Age of participants ranged from 19 to 82 years, with European participants significantly older ($p = 0.002$, Kruskal–Wallis). Previous infection was significantly less frequent in Europeans ($p = 0.005$, Chi$^2$ test). BMI was significantly lower in Europeans ($p < 0.001$, Kruskal–Wallis test). Levels of anti-S IgG were significantly higher ($p < 0.001$, Tables 1 and S2) and the ability to neutralize omicron was significantly more frequent ($p < 0.001$, S2 Table) 1 month after the third dose than after the second dose, irrespective of the infection status (Fig 3A). Previous SARS-CoV-2 infection was significantly associated with higher titers after the third dose ($p < 0.001$, Fig 3A). After the second dose, omicron neutralization was significantly more frequent in those with a previous infection ($p = 0.004$, S2 Table).

**Table 1. Factors associated with anti-S IgG levels 1 month after immunization (linear regression).**

| | N = 305 | Crude effect (95% CI) | p Value | Adjusted effect (95% CI) All variables | p Value | Adjusted effect (95% CI) Backward stepwise procedure | p Value |
|---|---|---|---|---|---|---|---|
| **Time point** | | | **<0.001** | | **<0.001** | | **<0.001** |
| Post second dose | 68 | *Reference* | | *Reference* | | *Reference* | |
| Post third dose | 237 | **+1.24 (1.02, 1.46)** | | **+1.28 (1.06, 1.51)** | | **+1.27 (1.06, 1.49)** | |
| **Infected** | | | **0.004** | | **<0.001** | | **<0.001** |
| No | 108 | *Reference* | | *Reference* | | *Reference* | |
| Yes | 197 | **+0.33 (0.11, 0.56)** | | **+0.36 (0.17, 0.55)** | | **+0.38 (0.19, 0.56)** | |
| **Gender**[a] | | | **0.051** | | **0.028** | | **0.036** |
| Male | 138 | **−0.18 (−0.36, 0.001)** | | **−0.20 (−0.39, −0.02)** | | **−0.19 (−0.37, −0.01)** | |
| Female | 167 | *Reference* | | *Reference* | | *Reference* | |
| **Age (years)**[a] | | | **0.046** | | 0.28 | | |
| 18–39 | 130 | *Reference* | | *Reference* | | | |
| 40–64 | 133 | −0.14 (−0.34, 0.06) | | −0.12 (−0.32, 0.09) | | | |
| ≥65 | 42 | **−0.35 (−0.63, −0.06)** | | −0.25 (−0.57, 0.07) | | | |
| **Comorbidities**[a] | | | 0.51 | | 0.70 | | |
| No | 173 | *Reference* | | *Reference* | | | |
| Yes | 132 | −0.06 (−0.25, 0.12) | | −0.04 (−0.24, 0.16) | | | |
| **BMI**[a] | | | **0.017** | | **0.014** | | **0.008** |
| Underweight | 9 | −0.26 (−0.80, 0.29) | | −0.51 (−1.09, 0.06) | | −0.48 (−1.05, 0.10) | |
| Normal | 93 | *Reference* | | *Reference* | | *Reference* | |
| Overweight | 90 | −0.04 (−0.27, 0.19) | | 0.02 (−0.21, 0.26) | | +0.05 (−0.28, 0.18) | |
| Obese | 113 | **+0.26 (0.04, 0.48)** | | **+0.27 (0.03, 0.50)** | | **+0.24 (0.03, 0.46)** | |
| **Community**[a] | | | 0.17 | | 0.34 | | |
| European | 79 | *Reference* | | *Reference* | | | |
| Melanesian | 55 | +0.02 (−0.26, 0.30) | | −0.11 (−0.40, 0.18) | | | |
| Polynesian | 55 | +0.24 (−0.04, 0.52) | | 0.03 (−0.27, 0.33) | | | |
| Other | 116 | +0.20 (−0.03, 0.44) | | 0.13 (−0.11, 0.36) | | | |

[a]The univariate analysis is adjusted for time point and infection status.

CI, confidence interval; BMI, body mass index.

BMI classes: Underweight = BMI <18.5 kg/m$^2$, Normal weight = BMI [18.5, 25[ kg/m$^2$, Overweight = BMI [25, 30[ kg/m$^2$, Obese = BMI ≥30 kg/m$^2$.

Statistically significant associations ($p < 0.05$) with anti-S IgG levels are shown in bold.

Given the high dependence of anti-S IgG levels on time point of analysis and infection status of the participants, the univariate analysis was adjusted for time point and infection status. Linear regression confirmed that number of doses and infection status were independently and significantly associated with higher anti-S IgG titers (adjusted effect +1.27, 95% CI (1.06, 1.49), $p < 0.001$ and +0.38, 95% CI (0.19, 0.56), $p < 0.001$, respectively, Table 1). Obesity was associated with significantly higher anti-S IgG titers (adjusted effect +0.24, 95% CI (+0.03, +0.46), $p = 0.008$). Male gender was significantly associated with lower anti-S IgG titers (adjusted effect −0.19, 95% CI (−0.37, −0.01), $p = 0.036$). Importantly, ethnic community was not associated with anti-S IgG titers ($p = 0.34$) (Fig 3B).

In a ROC curve analysis, a threshold of 5.737 AU for anti-S IgG titers maximized the proportion of subjects correctly classified in terms of neutralization capacity, but importantly the threshold maximized the capacity to identify those unable to neutralize omicron (57 out of 60, 95% specificity) (S3 Fig).

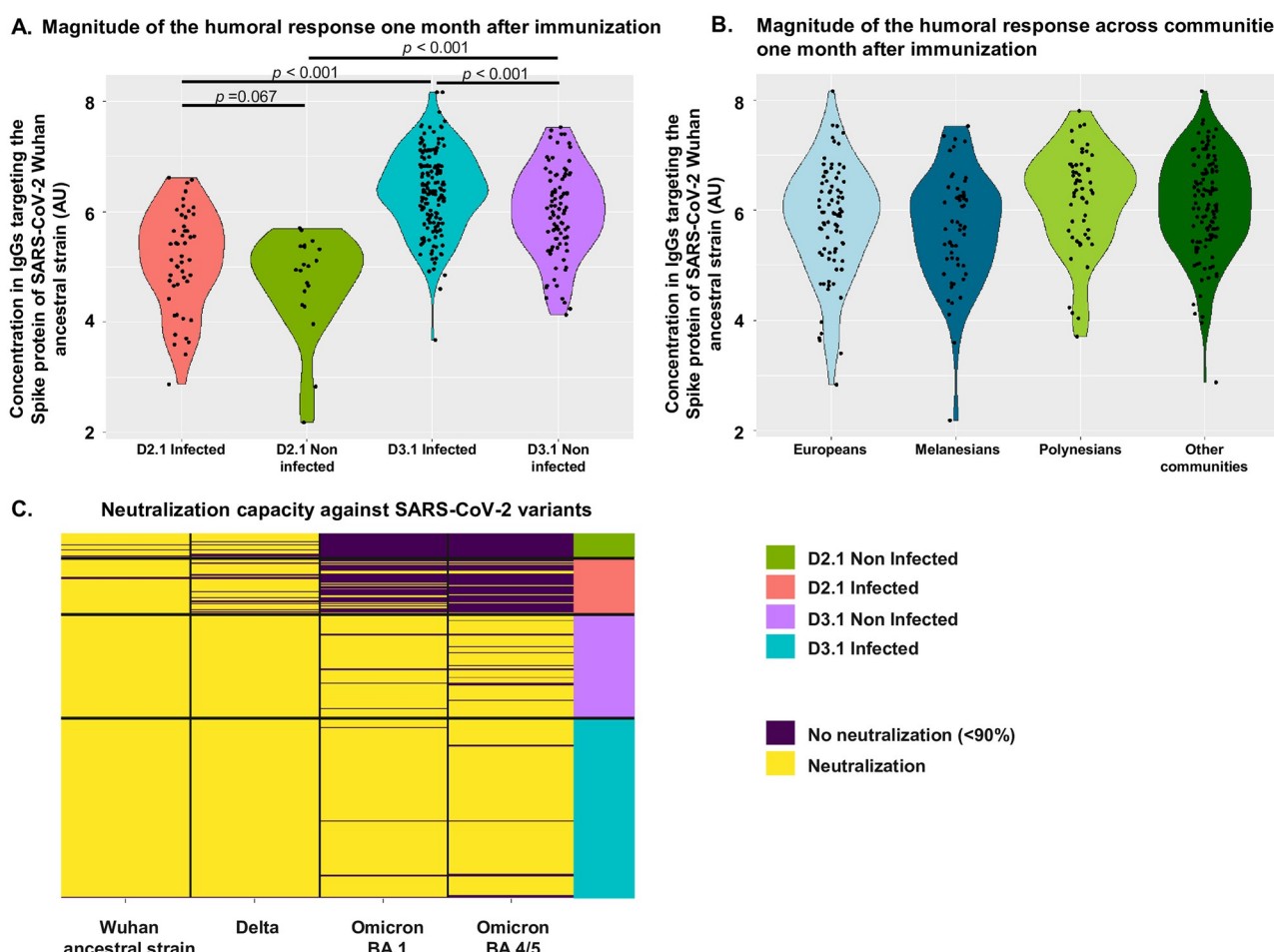

**Fig 3. Measure of anti-S IgG levels and neutralization capacity.** (A) Concentrations in IgG targeting the spike protein of SARS-CoV-2 ancestral strain Wuhan in samples collected 1 month after the second dose from previously infected participants (D2.1 infected, red), 1 month after the second dose from previously non-infected participants (D2.1 non-infected, green), 1 month after the third dose from previously infected participants (D3.1 infected, blue), and 1 month after the third dose from previously non-infected participants (D3.1 non-infected, purple). (B) Concentrations in IgG targeting the spike protein of SARS-CoV-2 ancestral strain Wuhan in samples collected from self-declared Melanesian (dark blue), European (light blue), Polynesian (light green) participants, or from participants self-declaring as belonging to other communities (dark green) 1 month after immunization. (C) Heatmap of the neutralization capacity against SARS-CoV-2 ancestral strain Wuhan, and against SARS-CoV-2 variants delta, omicron BA.1 and omicron BA.4/5, 1 month after the second dose in previously non-infected participants (D2.1 non-infected, green), 1 month after the second dose in previously infected participants (D2.1 infected, red), 1 month after the third dose in previously non-infected participants (D3.1 non-infected, violet), and 1 month after the third dose in previously infected participants (D3.1 infected, blue). Dark blue denotes the absence of neutralization capacity (<90%) and yellow denotes a neutralization capacity of the corresponding variant (≥90%) of 1/40 diluted sera. Each line represents 1 participant.

Pseudovirus neutralization assays showed that almost all individuals sampled 1 month after immunization were able to neutralize the vaccine strain Wuhan and the delta variant (Fig 3C). The ability to neutralize omicron BA.1 and/or BA.4/5, however, significantly differed by number of doses (Fig 3C): the proportion of participants neutralizing omicron increased from 19.1% after the second dose to 97.9% after the third dose (S2 Table). When considered separately, 17.6% (12/68) and 11.8% (8/68) of the participants were able to neutralize omicron BA.1 and omicron BA.4/5 at 1 month post-second dose, respectively. In contrast, 97.0% (230/237) and 93.7% (222/237) of the participants were able to neutralize omicron BA.1 and omicron BA.4/5 at 1 month post-third dose, respectively. For both variants, the proportion of neutralizing participants increased significantly between the 2 doses ($p < 0.0001$, Chi$^2$ test). In

addition, the proportion of participants neutralizing omicron BA.1 was not significantly different from the proportion of participants neutralizing omicron BA.4/5, neither at 1 month post-second dose ($p = 0.33$) nor at 1 month after the third dose ($p = 0.08$). In analyses stratified by immunization dose, only anti-S IgG levels were significantly associated with neutralization (OR (95% CI): 88.33 (15.79, 805.92), $p < 0.001$ at D2.1, and 12.00 (1.73, 237.30), $p = 0.028$ at D3.1 Table 2). After 2 immunizations, only anti-S IgG levels remained significantly associated with omicron neutralization ($p < 0.001$). After 3 immunizations, in univariate analysis, no factor other than anti-S IgG levels ($p = 0.028$) was associated with omicron neutralization (all $p$ values >0.20). Importantly, ethnic community was not associated with different odds to neutralize omicron ($p = 0.59$ and $p = 0.60$ at D2.1 and D3.1, respectively).

All participants but 7 were able to activate CD16 following exposure to SARS-CoV-2 spike, reflecting a capacity to mediate ADCC. Multivariate linear regression showed that anti-S IgG levels ≥5.737 AU and male gender were independently associated with higher ADCC activation (+0.61 95% CI (0.29, 0.94), $p < 0.001$ and +0.47 95% CI (0.14, 0.79), $p = 0.005$, respectively, Table 3). The ethnic community was not associated with the capacity to mediate ADCC ($p = 0.069$ in multivariate linear regression).

Sensitivity analyses not considering the "Other communities" group performed at 1-month post-immunization confirmed our initial findings regarding the lack of difference between the Melanesian, European, and Polynesian communities in terms of anti-S IgG levels, capacity to neutralize omicron and capacity to mediate ADCC (S3–S5 Tables).

## Humoral response to SARS-CoV-2 over 6 months post-third dose is similar across ethnic communities

Among the 214 participants who were evaluated at both D3.1 and D3.6, 29 self-declared as Melanesians, 57 as Europeans, 42 as Polynesians, and 86 as belonging to other communities (S6 Table). Europeans were significantly older ($p = 0.022$, Kruskal–Wallis test), showed less frequently signs of previous infection ($p = 0.059$, Chi$^2$ test) and had lower BMI ($p < 0.001$, Kruskal–Wallis test) (S6 Table). Between D3.1 and D3.6, anti-S IgG levels decreased in median (IQR) by 1.67 AU (0.63; 2.59) (S6 Table). The multivariate linear regression model showed that in participants never infected and with anti-S IgG levels <5.737 at D3.1, the anti-S IgG levels decreased (in mean (95% CI)) by 1.98 AU (1.56, 2.39) between D3.1 and D3.6 (S7 Table). In participants with anti-S IgG levels ≥5.737 at D3.1, the decrease was significantly steeper (−2.52 AU 95% CI (−2.90, −2.14), $p < 0.001$). In participants with breakthrough infection, anti-S IgG levels remained stable. In participants who were anti-N IgG positive at D3.1, anti-N IgG levels were independent from the ethnic community ($p$ value = 0.056, S8 Table). Among those participants, in those who did not show signs of reinfection (decrease or stable anti-N IgG level) at D3.6, the decrease in anti-N IgG was steeper in the Melanesian community (9 participants, $p$ value = 0.005, S9 Table).

Two hundred and nine participants out of 214 showed the ability to neutralize omicron at D3.1 (S6 Table). Of them, 35 (16.7%) had lost this ability by D3.6 (S6 Table). Multivariate logistic regression evidenced that previously infected individuals had significantly lower odds to lose omicron neutralization capacity (adjusted OR (95% CI): 0.20 (0.07, 0.53), 0.06 (0.01, 0.24), and 0.04 (0.01, 0.15) for previous infection, breakthrough infection and reinfection, respectively; $p < 0.001$, S10 Table). Males had higher odds to lose this ability (adjusted OR (95% CI): 2.64 (1.12, 6.56); $p = 0.030$, S10 Table). Compared to the participants aged <40 years, those aged 40 to 64 years and ≥65 years had higher odds to lose omicron neutralization capacity (adjusted OR (95% CI): 5.74 (1.87, 22.19) and 7.89 (2.02, 36.13), respectively;

**Table 2. Factors associated with the ability to neutralize omicron BA.1 and/or BA.4/5 one month after the second and third dose (logistic regression).**

*One month after the second dose*

| | N = 68 | Neutralization, n (%) | Crude OR (95% CI) | p Value | Adjusted OR (95% CI) All variables | p Value |
|---|---|---|---|---|---|---|
| **Infected**[a] | | | | **0.004** | | 0.20 |
| No | 21 | 0 (0) | 1 | | 1 | |
| Yes | 47 | 13 (27.7) | **16.83 (2.04, 2.19)** | | 6.30 (0.54, 848.08) | |
| **Level of anti-S IgG** | | | | **<0.001** | | **0.001** |
| <5.737 AU | 56 | 3 (5.4) | 1 | | 1 | |
| ≥ 5.737 AU | 12 | 10 (83.3) | **88.33 (15.79, 805.92)** | | **14.09 (2.73, 111.09)** | |
| **Gender** | | | | 0.60 | | 0.70 |
| Male | 31 | 5 (16.1) | 0.70 (0.19, 2.36) | | 0.68 (0.08, 3.84) | |
| Female | 37 | 8 (21.6) | 1 | | 1 | |
| **Age (years)**[a] | | | | 0.69 | | 0.62 |
| 18–39 | 42 | 9 (21.4) | 1 | | 1 | |
| 40–64 | 20 | 4 (20.0) | 0.96 (0.25, 3.30) | | 2.50 (0.28, 28.64) | |
| ≥65 | 6 | 0 (0) | 0.27 (0.00, 2.67) | | 1.96 (0.01, 88.14) | |
| **Comorbidities** | | | | 0.60 | | 0.50 |
| No | 38 | 8 (21.1) | 1 | | 1 | |
| Yes | 30 | 5 (16.7) | 0.75 (0.20, 2.54) | | 0.49 (0.06, 3.31) | |
| **BMI** | | | | 0.14 | | 0.49 |
| Normal weight | 20 | 2 (10) | 1 | | 1 | |
| Overweight | 23 | 3 (13.0) | 1.35 (0.20, 11.14) | | 1.75 (0.17, 21.39) | |
| Obese | 25 | 8 (32) | 4.24 (0.90, 30.81) | | 3.47 (0.33, 48.77) | |
| **Community** | | | | 0.59 | | 0.55 |
| European | 16 | 3 (18.8) | 1 | | 1 | |
| Melanesian | 22 | 3 (13.6) | 0.68 (0.11, 4.20) | | 0.45 (0.02, 6.94) | |
| Polynesian | 9 | 1 (11.1) | 0.54 (0.02, 5.11) | | 0.37 (0.01, 9.72) | |
| Other | 21 | 6 (28.6) | 1.73 (0.38, 9.54) | | 1.54 (0.14, 19.80) | |

*One month after the third dose*

| | N = 237 | Neutralization, n (%) | Crude OR (95% CI) | p Value | Adjusted OR (95% CI) All variables | p Value |
|---|---|---|---|---|---|---|
| **Infected** | | | | 0.30 | | 0.60 |
| No | 87 | 84 (96.6) | 1 | | 1 | |
| Yes | 150 | 148 (98.7) | 2.64 (0.43, 20.36) | | 1.63 (0.29, 9.90) | |
| **Level of anti-S IgG** | | | | **0.028** | | **0.038** |
| <5.737 AU | 62 | 58 (93.5) | **1** | | **1** | |
| ≥ 5.737 AU | 175 | 174 (99.4) | **12.00 (1.73, 237.30)** | | **6.78 (1.11, 74.79)** | |
| **Gender** | | | | 0.50 | | 0.30 |
| Male | 130 | 128 (98.5) | 0.54 (0.07, 3.33) | | 0.35 (0.04, 2.25) | |
| Female | 107 | 104 (97.2) | 1 | | 1 | |
| **Age (years)**[a] | | | | 0.40 | | 0.30 |
| 18–39 | 88 | 88 (100) | 1 | | 1 | |
| 40–64 | 113 | 109 (96.5) | 0.14 (0.00, 1.32) | | 0.20 (0.00, 2.23) | |
| ≥65 | 36 | 35 (97.2) | 0.13 (0.00, 2.57) | | 0.45 (0.00, 22.47) | |
| **Comorbidities** | | | | 0.40 | | 0.50 |
| No | 135 | 133 (98.5) | 1 | | 1 | |
| Yes | 102 | 99 (97.1) | 0.50 (0.06, 3.05) | | 0.55 (0.09, 3.29) | |
| **BMI** | | | | 0.32 | | 0.099 |
| Underweight | 8 | 7 (87.5) | 0.19 (0.02, 4.49) | | 0.19 (0.01, 3.14) | |

*(Continued)*

**Table 2.** (Continued)

| | | | | | | |
|---|---|---|---|---|---|---|
| Normal weight | 74 | 72 (97.3) | 1 | | 1 | |
| Overweight | 67 | 66 (98.5) | 1.83 (0.17, 40.00) | | 2.74 (0.32, 36.58) | |
| Obese | 88 | 87 (98.9) | 2.42 (0.23, 52.63) | | 4.87 (0.36, 131.9) | |
| Community[a] | | | | 0.60 | | 0.31 |
| European | 63 | 61 (96.8) | 1 | | 1 | |
| Melanesian | 33 | 33 (100) | 2.72 (0.21, 379.95) | | 1.00 (0.05, 150.52) | |
| Polynesian | 46 | 44 (95.7) | 0.72 (0.11, 4.85) | | 0.20 (0.01, 2.43) | |
| Other | 95 | 94 (98.9) | 2.56 (0.33, 28.48) | | 1.05 (0.11, 11.72) | |

CI, confidence interval; BMI, body mass index; OR, odds ratio.

[a]Logistic regression with Firth's correction.

BMI classes: Underweight = BMI $<18.5$ kg/m$^2$, Normal weight = BMI [18.5, 25[ kg/m$^2$, Overweight = BMI [25, 30[ kg/m$^2$, Obese = BMI $\geq 30$ kg/m$^2$.

Statistically significant associations ($p < 0.05$) with the ability to neutralize omicron BA.1 and/or BA.4/5 are shown in bold.

$p = 0.008$, S10 Table). Importantly, ethnic community was not associated with the loss of omicron neutralization capacity between D3.1 and D3.6 (S10 Table).

ADCC activation remained stable between D3.1 and D3.6 (median (95% CI) difference (+0.20 AU ($-0.41$, 0.61)). The multivariate linear regression model showed that the capacity to mediate ADCC in female participants who were never infected remained stable in time (mean (95% CI) progression $-0.16$ ($-0.53$, 0.21)) (S11 Table). In participants with breakthrough infection between D3.1 and D3.6, the capacity to mediate ADCC significantly increased between D3.1 and D3.6 (mean (95% CI) progression of +0.51 AU (0.13, 0.88), $p = 0.0087$). In male participants, the capacity to mediate ADCC significantly decreased between D3.1 and D3.6 (mean (95% CI) progression of $-0.47$ AU ($-0.85$, $-0.08$), $p < 0.018$)). Importantly, the ethnic community was not associated with an evolution in the capacity to mediate ADCC ($p = 0.80$ in univariate linear regression).

## Humoral response to SARS-CoV-2 at 6 months post-third dose is similar across ethnic communities

Among the 488 participants sampled at D3.6, 119 self-declared as Melanesians, 166 as Europeans, 68 as Polynesians, and 135 belonged to other communities (S12 Table). Europeans were older, had lower BMIs, exhibited signs of previous infection less frequently, and were less able to neutralize omicron. Past infection was significantly more frequent in Melanesians and "other communities" ($p < 0.001$, Chi$^2$ test). Previous infection was significantly associated with higher anti-S IgG levels (crude effect +1.16, 95% CI (0.91, 1.41), $p < 0.001$, S13 Table). Melanesians and other communities showed higher anti-S IgG levels in univariate analysis. In multivariate analysis, only the infection status remained significantly associated with higher anti-S IgG levels.

Of these participants, 362 (74.2%) showed ability to neutralize omicron. In the multivariate model, previous infection and higher anti-S IgG levels were independently associated with higher odds of neutralization at D3.6 (adjusted OR (95% CI): 4.04 (2.54, 6.50), $p < 0.001$ and 51.64 (11.00, 922.94), $p < 0.001$, respectively; S14 Table). Conversely, males had lower odds to neutralize omicron (adjusted OR (95% CI): 0.61 (0.38, 0.97), $p = 0.024$). As compared to participants aged <45 years, those aged 40 to 64 years and $\geq 65$ years also had lower odds to neutralize omicron (adjusted OR (95% CI): 0.48 (0.24, 0.90) and 0.29

**Table 3. Factors associated with the level of ADCC (CD16 activation in response to SARS-CoV-2 spike protein from the ancestral strain Wuhan) 1 month after immunization (linear regression).**

| | n = 303 | Crude effect (95% CI) | p Value | Adjusted effect (95% CI) All variables | p Value | Adjusted effect (95% CI) Backward stepwise procedure | p Value |
|---|---|---|---|---|---|---|---|
| **Time point** | | | 0.10 | | 0.90 | | |
| Post second dose | 66 | *Reference* | | *Reference* | | | |
| Post third dose | 237 | +0.33 (−0.06, 0.73) | | −0.02 (−0.48, 0.44) | | | |
| **Infected** | | | 0.30 | | 0.40 | | |
| No | 108 | *Reference* | | *Reference* | | | |
| Yes | 195 | +0.18 (−0.16, 0.53) | | 0.16 (−0.19, 0.51) | | | |
| **Level of anti-S IgG** | | | **<0.001** | | **0.003** | | **<0.001** |
| <5.737 AU | 117 | *Reference* | | *Reference* | | *Reference* | |
| ≥ 5.737 AU | 186 | **+0.58 (0.25, 0.92)** | | **0.60 (0.21, 0.99)** | | **+0.61 (0.29, 0.94)** | |
| **Gender** | | | **0.010** | | **0.011** | | **0.005** |
| Male | 136 | **+0.43 (0.10, 0.76)** | | **0.42 (0.10, 0.75)** | | **+0.47 (0.14, 0.79)** | |
| Female | 167 | *Reference* | | *Reference* | | *Reference* | |
| **Age (years)** | | | 0.48 | | 0.32 | | |
| 18–39 | 129 | *Reference* | | *Reference* | | | |
| 40–64 | 132 | 0.13 (−0.22, 0.48) | | 0.08 (−0.29, 0.45) | | | |
| ≥65 | 42 | −0.17 (−0.68, 0.34) | | −0.32 (−0.89, 0.25) | | | |
| **Comorbidities** | | | 0.60 | | 0.90 | | |
| No | 172 | *Reference* | | *Reference* | | | |
| Yes | 131 | −0.10 (−0.43, 0.23) | | −0.02 (−0.39, 0.34) | | | |
| **BMI** | | | 0.25 | | 0.19 | | |
| Underweight | 8 | −1.06 (−2.11, −0.01) | | −1.04 (−2.08, −0.01) | | | |
| Normal | 94 | *Reference* | | *Reference* | | | |
| Overweight | 89 | −0.07 (−0.49, 0.35) | | 0.12 (−0.31, 0.54) | | | |
| Obese | 113 | −0.14 (−0.54, 0.26) | | −0.03 (−0.45, 0.40) | | | |
| **Community** | | | **0.05** | | 0.051 | | |
| European | 78 | *Reference* | | *Reference* | | | |
| Melanesian | 54 | **−0.71 (−1.21; −0.20)** | | −0.69 (−1.21, −0.17) | | | |
| Polynesian | 55 | **−0.36 (−0.86; 0.14)** | | −0.50 (−1.03, 0.04) | | | |
| Other | 116 | **−0.35 (−0.76; 0.07)** | | −0.45 (−0.87, −0.03) | | | |

CI, confidence interval; BMI, body mass index.

BMI classes: Underweight = BMI <18.5 kg/m$^2$, Normal weight = BMI [18.5, 25[ kg/m$^2$, Overweight = BMI [25, 30[ kg/m$^2$, Obese = BMI ≥30 kg/m$^2$.

Statistically significant associations ($p < 0.05$) with ADCC levels are shown in bold.

(0.14, 0.57), $p = 0.0024$). Capacity to neutralize omicron at D3.6 was independent of the ethnic community.

## Discussion

This study found that Europeans, Melanesians, Polynesians, and other communities had equivalent anti-S IgG titers and capacity to neutralize omicron and mediate ADCC following Pfizer BNT162b2 vaccination and SARS-CoV-2 infection. They showed a similar maintenance of their hybrid humoral response 6 months after immunization.

By assessing the hybrid humoral immune response in a large cohort of 585 participants from New Caledonia belonging to the Melanesian, Polynesian, European, or other communities, the current study fills a knowledge gap in the characterization of Pacific Islanders'

humoral immune response up to 6 months after a third dose of Pfizer BNT162b2 vaccine. We report an unprecedented characterization of anti-S IgG levels but also the functionality of these antibodies (capacity to neutralize omicron BA.1 and/or BA.4/5 and to mediate ADCC) in an underrepresented population. Melanesians' and other communities' higher anti-S IgG levels at D3.6 were associated with more frequent infections, probably resulting from different lifestyles across communities in New Caledonia, which may favor transmission [2]. The higher SARS-CoV-2 transmission we observed in these communities is in line with the higher COVID-19 seroprevalence detected among Oceanians in New Caledonia [18]. Our results are also consistent with the robust antibody response and neutralization observed after 2 doses of BNT162b2 vaccine in a cohort comprising 80 Pacific people [11].

Population genetic studies have revealed that ancient admixture between humans and Denisovans left traces in present-day Pacific Islanders' genomes [19], introducing genetic variants almost exclusively related to immune response regulation [20]. How genetic traits from archaic humans contribute to phenotypic differences between contemporary individuals, especially relating to their immune response, is poorly known. We showed here that Pacific Islanders' genetic uniqueness did not translate in differential hybrid humoral response to SARS-CoV-2. Although some genetic variants have been associated with antigen-specific IgG levels [21], neither their presence in Pacific populations nor their impact on anti-SARS-CoV-2 IgG levels have been explored.

We found that hybrid immunity resulting from BNT162b2 vaccination and SARS-CoV-2 infection was highly prevalent in our cohort. The infecting serotype was not determined in the framework of the current study. However, due to COVID-19 epidemiology in New Caledonia (S1 Fig), it is possible to infer that while participants with a history of past infection sampled at D2.1 were likely infected by the delta variant, participants with a history of past infection sampled at later time points could have been infected either by the delta or omicron variants. In our study, hybrid immunity translated in a robust anti-S IgG secretion with efficient omicron neutralization capacity compared to noninfected vaccinees, in Pacific Islanders as well. We evidenced that anti-S IgG levels above 5.737 AU in LuLISA assay discriminate with 95% specificity the participants able to neutralize omicron, providing useful decision-aiding tool to accurately identify those likely to benefit from a booster dose. Despite a steeper decrease in anti-N IgG over time in the Melanesian community, albeit in a reduced number of participants, we show that the humoral immune response to natural infection was overall similar across communities. A third dose of BNT162b2 vaccine resulted in omicron neutralization in almost all participants, independently from a previous infection. This is in line with the acquisition by memory B cells of the capacity to recognize omicron BA.2 and BA.5 variants after a booster dose of BNT162b2 [22]. Consistently, hybrid immunity was shown to boost antibody levels and poly-functionality regardless of the sequence of natural infection and vaccination [23], resulting in a significant and lasting protection against omicron infection [24]. Recall of memory B cells generated after the original infection contributes to this increase in variant-neutralizing antibodies after vaccination [25]. Enhancement of variant-neutralizing capacity in infected vaccinees supports the relevance to vaccinate Pacific Islanders, who exhibit higher COVID-19 seroprevalence [18].

We observed that self-declared males had lower anti-S IgG titers at 1 month after immunization, which did not translate in differential neutralization capacity. Sex did not affect antibody decline between D3.1 and D3.6. However, the loss of omicron neutralization capacity between D3.1 and D3.6 was more likely in males, resulting in lower odds of omicron neutralization at D3.6. Furthermore, the capacity to mediate ADCC was enhanced at 1-month post-immmunization but decreased more rapidly in males. Consistently, males were shown to secrete lower antibody levels after 2 doses of BNT162b2 [26]. However, anti-S IgG levels were

shown to decline faster in males after infection [27]. Higher antibody titers in females were observed for several vaccines [28,29], suggesting a differential effective vaccine dose according to sex. Overall, while several studies demonstrated an impact of sex on the humoral response [27,30], the underlying mechanisms remain poorly characterized.

In addition, we found that age did not affect antibody decline between D3.1 and D3.6, probably due to significantly lower anti-S IgG levels in the ≥40 years age groups at 1-month after immunization. Consistently, older individuals exhibit lower antibody levels after 2 doses of BNT162b2, with similar decline across ages [26]. Individuals aged ≥40 years had lower odds of omicron neutralization at D3.6. Age might therefore impact humoral response polyfunctionality. SARS-CoV-2 vaccination's outcome correlates with signs of immunosenescence [31], probably contributing to this loss of omicron neutralization in older individuals. This highlights the importance to offer booster doses to the elderly to prevent the loss of omicron neutralization capacity, in Pacific Islanders as well.

In this study, higher BMIs were associated with higher anti-S IgG levels at 1-month post-immunization, in turn associated with higher odds of omicron neutralization. Although excess adiposity is reported to negatively impact immune function and host defense in obese individuals [32], presence of visceral pro-inflammatory adipose tissues [33], dysregulation in extracellular vesicles derived from adipose tissue and imbalance in the biosynthesis and levels of polyunsaturated fatty acid-derived oxylipins [34] as well as secretion of adipokines and hypoxia [35] in obese individuals maintain a basal pro-inflammatory state prone to activate immune cells, which may contribute to an enhanced immune response, including the early secretion of higher antibody levels. However, impact of BMI on humoral immunity to natural infection and BNT162b2 vaccination is controversial [17,36–38]. Nonetheless, this result emphasizes the benefits of vaccinating obese individuals, who are overrepresented among Pacific Islanders and at risk for severe COVID-19 [39].

The strengths of the current study are: (i) the considerable dataset of reliable clinico-epidemiological and biological data collected in a large cohort of Pacific Islanders, which is unlikely to be available elsewhere; (ii) the comprehensive characterization of their humoral immune response, including anti-S and anti-N IgG levels in LuLISA assays but also omicron neutralization capacity in pseudovirus neutralization assay against omicron BA.1 and BA.4/5 and the capacity to mediate ADCC in CD16 activation assays; (iii) community engagement and population acceptance of a study on Pacific Islanders' singularities, which paves the way to tailored health research and public health interventions in an underrepresented population.

The current study suffers limitations: first, although the LuLISA assay quantified total anti-spike IgG levels and not variant-specific IgG levels, the pseudovirus neutralization assays allowed to discriminate sera containing variant-specific anti-S IgG levels above the neutralizing threshold. Second, the high rate of SARS-CoV-2 infections prevented us from being able to assess ethnicity's impact on the humoral response to vaccination alone. Nonetheless, infection status was taken into consideration in multivariate models. As SARS-CoV-2 immunity worldwide is predominantly hybrid, characterization of a population's immunity, independently of its infection status, is highly relevant to adapt Public Health strategies. Third, the current study relied on self-declared ethnicity, without genetic determination of ancestry. Although mixed marriages are common, participants had the opportunity to identify as "Other communities," which included people of mixed race. Self-declared ethnicity was thus an appropriate surrogate to genetically determined ancestry. The "Other communities" group was thus, as per definition, a highly diverse group of participants rich from its various genetic ancestries. Community engagement and population acceptance were paramount for both the successful implementation of the current study and subsequent ownership taking of the results by New Caledonia's population. The current study was therefore inclusive and enrolled any participant regardless

of her/his self-declared ethnicity, including people of non-European, non-Melanesian, or non-Polynesian ancestry. Finally, the sample size was not reached for the Polynesian community. Under-recruitment tends to bias the results towards non significance; therefore, one cannot exclude that this could explain the similar responses to vaccination we observed in the different communities. However, the lack of community effect, despite different prevalence of obesity, suggests that SARS-CoV-2 hybrid humoral response does not differ between communities.

This study has important implications: evidence of a robust and prototypical humoral immune response to Pfizer BNT162b2 vaccine and SARS-CoV-2 in Pacific Islanders is of prime importance to foster booster doses roll-out, with the aim to better protect this under-studied population, at risk for severe COVID-19. Furthermore, as obesity is highly prevalent in Pacific populations and is associated with higher anti-spike antibody levels, results from this study will contribute to adapted vaccination regimens, including booster doses roll-out, to better protect Pacific Islanders. This study focused on the humoral immune response. Singularities of Pacific Islanders' innate and cellular immune responses elicited by SARS-CoV-2 infection and BNT162b2 vaccination, and their potential contribution to the peculiar epidemiology of COVID-19 in Pacific Islanders communities remain to be assessed to build a more holistic view of Pacific Islanders' response to COVID-19 vaccination.

In conclusion, as New Caledonia recapitulates the ethnic diversity of the South Pacific region, the current study provides pivotal knowledge on the immune response of any Pacific Islander to SARS-CoV-2, to the benefit of all inhabitants from the South Pacific region. Overall, by investigating the hybrid immune response of Pacific Islanders to SARS-CoV-2, our study sheds light on the health of a population neglected in human studies, opening the door to considering Pacific Islanders' uniqueness in Health studies.

## Supporting information

**S1 Fig. Distribution of samples according to COVID-19 epidemiology in New Caledonia.** Absolute number of cases are shown in black on the left y axis, with delta, omicron BA.1 and omicron BA.4/5 epidemic peaks. Absolute number of samples collected 1 month after the second dose (orange), 3 months after the second dose (green), 1 month after the third dose (blue), or 6 months after the third dose (violet) are shown on the right y axis. Inclusion periods are shown below the graph.
(PDF)

**S2 Fig. Distribution of the communities among the 585 sampled participants, according to the enrolment date.**
(TIF)

**S3 Fig. ROC curve for the anti-S IgG threshold best discriminating samples exhibiting omicron neutralization capacity.**
(TIF)

**S1 Table. Participants' description at baseline (1 month after the second or third dose).**
(DOCX)

**S2 Table. Comparison of immune characteristics after the second or third dose.**
(DOCX)

**S3 Table. Factors associated with anti-S IgG levels 1 month after immunization, not considering participants from "Other communities" (linear regression).**
(DOCX)

**S4 Table. Factors associated with the ability to neutralize omicron BA.1 and/or BA.4/5 one month after the second and third dose, not considering participants from "Other communities" (logistic regression).**
(DOCX)

**S5 Table. Factors associated with the level of ADCC (CD16 activation) 1 month after immunization, not considering participants from "Other communities" (linear regression).**
(DOCX)

**S6 Table. Description of the participants followed up at month 1 and month 6 post-third dose of vaccination (N = 214).**
(DOCX)

**S7 Table. Factors associated with the progression of anti-S IgG levels between 1 and 6 months after immunization (linear regression).**
(DOCX)

**S8 Table. Impact of ethnicity on the levels of anti-N IgG at 1-month post-immunization in infected individuals (linear regression).**
(DOCX)

**S9 Table. Impact of ethnicity on the decrease in the levels of anti-N IgG between 1 and 6 months post-third dose in individuals infected at 1 month post-third dose (linear regression).**
(DOCX)

**S10 Table. Factors associated with the loss of the ability to neutralize omicron BA.1 and/or BA.4/5 between 1 and 6 months after immunization (logistic regression).**
(DOCX)

**S11 Table. Factors associated with the variation in the level of ADCC (CD16 activation in response to SARS-CoV-2 spike protein from the ancestral strain Wuhan) between 1 and 6 months after immunization (linear regression).**
(DOCX)

**S12 Table. Description of participants sampled at 6 months after the third dose of immunization.**
(DOCX)

**S13 Table. Factors associated with the levels of anti-S IgG 6 months after the third dose (linear regression).**
(DOCX)

**S14 Table. Factors associated with the ability to neutralize omicron BA.1 and/or BA.4/5 six months after the third dose (logistic regression).**
(DOCX)

**S1 Data. Data sets for participants sampled at 1-month post-immunization (1 month), participants followed up between 1 and 6 months post-third dose (D3.1-D3.6) and participants sampled at 6 months post-third dose (D3.6).**
(XLSX)

**S1 Protocol. Extended synopsis of the study protocol.**
(PDF)

**S1 STROBE Checklist. STROBE Statement—Checklist of items that should be included in reports of cohort studies.**
(DOCX)

## Acknowledgments

We would like to thank all the participants to the COVCAL study. This study has been labeled as a National Research Priority by the National Orientation Committee for Therapeutic Trials and other researches on COVID-19 (CAPNET). The investigators would like to acknowledge ANRS | Emerging infectious diseases for its scientific support and funding, the French Ministry of Health and Prevention and the French Ministry of Higher Education, Research and Innovation for their funding and support. We warmly thank the Clinical Research Department of the Centre for Translational Research at Institut Pasteur in Paris, and especially Dr. Nathalie Clément, for their support in ethic procedures. The authors would also like to thank Priscillia Piersanti, Aurore Martini, Sabine Crescentini, Dr. Christophe Assié, and Dr. Johanna Read along with home-visiting nurses and personnel from Medical Centers for their support in the investigation; Stéphane Petres and the Plateforme Technologique de Production et Purification de Protéines Recombinantes at Institut Pasteur for the production of the N and S proteins used in the LuLISA assays; Faustine Amara, Margot Penru, Yannis Rahou, and Lou-Léna Vrignaud for technical support; and Dr. Etienne Simon-Lorière, Pr Arnaud Fontanet, and Pr John Aaskov for insightful discussions.

## Author Contributions

**Conceptualization:** Catherine Inizan, Adrien Courtot, Thibaut Demaneuf, Georges Médevielle, Marc Jouan, Valérie Albert-Dunais, Myrielle Dupont-Rouzeyrol.

**Data curation:** Catherine Inizan, Adrien Courtot, Chloé Sturmach, Timothée Bruel, Sylvie van der Werf.

**Formal analysis:** Catherine Inizan, Adrien Courtot, Chloé Sturmach, Thibaut Demaneuf, Sylvie van der Werf, Yoann Madec.

**Funding acquisition:** Catherine Inizan, Georges Médevielle, Marc Jouan, Valérie Albert-Dunais, Myrielle Dupont-Rouzeyrol.

**Investigation:** Catherine Inizan, Adrien Courtot, Chloé Sturmach, Anne-Fleur Griffon, Antoine Biron, Timothée Bruel, Vincent Enouf, Olivier Schwartz, Ann-Claire Gourinat, Georges Médevielle, Marc Jouan, Sylvie van der Werf, Valérie Albert-Dunais, Myrielle Dupont-Rouzeyrol.

**Methodology:** Catherine Inizan, Thibaut Demaneuf, Georges Médevielle, Sylvie van der Werf, Yoann Madec, Valérie Albert-Dunais, Myrielle Dupont-Rouzeyrol.

**Project administration:** Catherine Inizan, Anne-Fleur Griffon, Georges Médevielle, Valérie Albert-Dunais, Myrielle Dupont-Rouzeyrol.

**Resources:** Catherine Inizan, Adrien Courtot, Anne-Fleur Griffon, Antoine Biron, Timothée Bruel, Sandie Munier, Olivier Schwartz, Georges Médevielle, Sylvie van der Werf, Valérie Albert-Dunais, Myrielle Dupont-Rouzeyrol.

**Supervision:** Georges Médevielle, Marc Jouan, Yoann Madec, Valérie Albert-Dunais, Myrielle Dupont-Rouzeyrol.

**Validation:** Catherine Inizan, Adrien Courtot, Yoann Madec, Myrielle Dupont-Rouzeyrol.

**Writing – original draft:** Catherine Inizan, Yoann Madec, Myrielle Dupont-Rouzeyrol.

**Writing – review & editing:** Adrien Courtot, Chloé Sturmach, Anne-Fleur Griffon, Antoine Biron, Timothée Bruel, Vincent Enouf, Thibaut Demaneuf, Sandie Munier, Olivier Schwartz, Ann-Claire Gourinat, Georges Médevielle, Marc Jouan, Sylvie van der Werf, Valérie Albert-Dunais.

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
