## [Editor Report · Decision Letter 0]

5 Apr 2024

Dear Dr Inizan, 

Thank you for submitting your manuscript entitled "Hybrid humoral immune response of Pacific Islanders to BNT162b2 vaccination and Delta/Omicron infection: a cohort study" for consideration by PLOS Medicine.

Your manuscript has now been evaluated by the PLOS Medicine editorial staff and I am writing to let you know that we would like to send your submission out for external peer review.

Please re-submit your manuscript within two working days, i.e. by Tuesday 9th April. Of course, please do let me know if you need more time.

Feel free to email the support team at plosmedicine@plos.org if you have any queries relating to your submission.

Kind regards,

Syba Sunny, MBBS, MRes, FRCPath

Associate Editor

PLOS Medicine

ssunny@plos.org

---

## [Decision Letter · Decision Letter 1]

5 Jun 2024

Dear Dr. Inizan,

Many thanks for submitting your manuscript "Hybrid humoral immune response of Pacific Islanders to BNT162b2 vaccination and Delta/Omicron infection: a cohort study" (PMEDICINE-D-24-01093R1) for consideration at PLOS Medicine. 

Your paper has now been reviewed by 3 subject reviewers and a statistician; the comments are included below. It was also discussed with an academic editor with relevant expertise and the wider editorial team. 

As you will see, the reviewers were largely positive about the paper. However, there were requests for further information and clarification on key aspects of the study. As such, I’m pleased to invite you to revise the paper in response to the reviewers’ comments and our editorial requests (see below). We plan to send the revised paper to some or all of the original reviewers*, and of course we cannot provide any guarantees at this stage regarding publication.

When you upload your revision, please include a point-by-point response that addresses all of the reviewer and editorial points, indicating the changes made in the manuscript and either an excerpt of the revised text or the location (e.g. page and line number) where each change can be found. Please submit a clean version of the paper as the main article file and a version with changes marked should as a marked-up manuscript. Please also check the guidelines for revised papers at http://journals.plos.org/plosmedicine/s/revising-your-manuscript for any that apply to your paper.

We ask that you submit your revision by Jun 26 2024 11:59PM. However, if this deadline is not feasible, please contact me by email (ssunny@plos.org), and we can discuss a suitable alternative.

Please don’t hesitate to contact me directly with any questions (ssunny@plos.org). 

Kind regards,

Syba

Syba Sunny MBBS, MRes, FRCPath

Associate Editor 

PLOS Medicine

ssunny@plos.org

*Please note: If your article is accepted, you may have the opportunity to make the peer review history publicly available. The record will include editor decision letters (with reviews) and your responses to reviewer comments. If eligible, we will contact you to opt in or out.

EDITORIAL COMMENTS:

The editors agree that the authors should be commended for performing valuable work with a population that has been under-represented. We are very grateful that you chose to submit your research for evaluation to PLOS Medicine. However, we do agree with the points raised by reviewers 1, 3 and 4, and ask that the authors respond to these in full. 

1) Data Availability:

The Data Availability Statement (DAS) requires revision. Given the data are not freely available, please describe briefly the ethical, legal, or contractual restriction that prevents you from sharing it. Please also include an appropriate contact (web or email address) for inquiries (but note, this cannot be a study author). Please see the following page for further information and guidance: 

https://journals.plos.org/plosmedicine/s/data-availability#loc-unacceptable-data-access-restrictions

2) Reporting guidance:

Please report your study according to the relevant guidance which can be found here https://www.equator-network.org/reporting-guidelines/

3) Statistical reporting: 

(Note that not all of the instructions below may be applicable to your work.)

Please quantify the main results with 95% CIs and p values.

When reporting p values please report as <0.001 and where higher as p=0.002, for example. When reporting 95% CIs please separate upper and lower bounds with commas instead of hyphens as the latter can be confused with reporting of negative values.

Please include the actual amounts and/or absolute risk(s) of relevant outcomes (including NNT or NNH where appropriate), not just relative risks or correlation coefficients. (example for absolute risks: PMID: 28399126).

Please include any important dependent variables that are adjusted for in the analyses.

4) Prespecified analysis plan/study protocol:

I note that you have included a protocol in the supporting information. Can you provide an English translation of this? Please also state that you had a prospective study protocol early in the Methods section. Also, note that any changes in the analysis-- including those made in response to peer review comments-- should be identified as such in the Methods section of the paper, with rationale.

5) Abstract layout:

Please structure your abstract using the PLOS Medicine headings (Background, Methods and Findings, Conclusions).

6) Author summary:

At this stage, we ask that you include a short, non-technical Author Summary of your research to make findings accessible to a wide audience that includes both scientists and non-scientists. The authors summary should consist of 2-3 succinct bullet points under each of the following headings:

• Why Was This Study Done? Authors should reflect on what was known about the topic before the research was published and why the research was needed.

• What Did the Researchers Do and Find? Authors should briefly describe the study design that was used and the study’s major findings. Do include the headline numbers from the study, such as the sample size and key findings. 

• What Do These Findings Mean? Authors should reflect on the new knowledge generated by the research and the implications for practice, research, policy, or public health. Authors should also consider how the interpretation of the study’s findings may be affected by the study limitations. In the final bullet point of ‘What Do These Findings Mean?’, please describe the main limitations of the study in non-technical language.

The Author Summary should immediately follow the Abstract in your revised manuscript. This text is subject to editorial change and should be distinct from the scientific abstract. Please see our author guidelines for more information: https://journals.plos.org/plosmedicine/s/revising-your-manuscript#loc-author-summary

7) Introduction layout:

Please address past research and explain the need for and potential importance of your study. If there has been a systematic review of the evidence related to your study (or you have conducted one), please refer to and reference that review and indicate whether it supports the need for your study.

8) Methods section:

Please move your Methods section so that it appears after your Introduction and before your Results in the main text.

9) Discussion layout:

Thank you for starting your Discussion with a short summary of the article's findings. For the result of the Discussion section, please present the following points in this order: what the study adds to existing research and where and why the results may differ from previous research; strengths and limitations of the study; implications and next steps for research, clinical practice, and/or public policy; one-paragraph conclusion.

COMMENTS FROM REVIEWERS:

Reviewer #1: This manuscript represents an important addition to the literature by presenting information on vaccine efficacy of an underrepresented population. The report is missing crucial information regarding the protection of this population by antibodies.

Minor comments:

- Was the study undertaken in line with the Declaration of Helsinki? If so, please specify

Major comments:

- Were the spike and nucleoprotein-specific responses both IgG and neutralisation measured in this study towards the Ancestral Wuhan virus? Please specify strain in methods section and on figures.

- Include information regarding the demographics of the study population including age and gender in first results paragraph

- If available include information about thich virus variant wave participants were infected in

- What was the timing between vaccine dose as this can have an impact on the magnitude of response measured.

- The authors note that neutralisation levels of Omicron is reduced 6 months after dose 6. What Omicron variant was this too? It is important to include data on the anti-Omicron spike IgG levels. 1. Virus-specific IgG levels will start to wane by 6 months post dose 3. 2. In the literature recognition of Omicron variants is significantly decreased see Hartley et al. 2023 BioRxIV (https://www.biorxiv.org/content/10.1101/2023.09.15.557929v1)

- There is no mention that other metrics of immunity should be measured to ensure a more holistic overview of the reponse of this population to vaccination

Reviewer #2: The manuscript assesses humoral vaccine responses to BNT162b2 vaccination in a cohort of participant from New Caledonia. 

I congratulate the authors on a good study with a large hard to collect and complicated set of data. The study is well conducted and findings are robust and of interest and importance to the research community.

However I do not think the findings "provide a substantial advance over existing knowledge". 

Only a few of the findings add to excising knowledge and whilst these are important, are not substantial advances.

Reviewer #3: The manuscript by Inizan et al titled 'Hybrid humoral immune response of Pacific Islanders to BNT162b2 vaccination and Delta/Omicron infection: a cohort study' is exceptionally well-written. It provides evidence from Pacific communities that the Pfizer mRNA vaccine is effective in elucidating humoral immune response and ADCC among adults and elderly people although the outcomes shown to vary across age and by sex. However, I have the following few comments for authors to address before their manuscript is being considered for publication in PLOS Medicine. 

-In addition to characterizing the humoral immune response of Pacific Islanders to the Pfizer BNT162b2 vaccine and SARS-CoV-2 infection, the study also investigated ADCC. I, therefore, suggest the authors consider modifying the title of their manuscript in such a way it also includes ADCC.

-Obesity is considered to be one of the comorbidities reported to be associated with developing symptomatic SARS-CoV-2 infection and severe COVID-19. Surprisingly, the current study observed that obese individuals (with a BMI > 30 kg/m²) had significantly higher anti-Spike IgG levels at one month post-immunization. This finding is intriguing and warrants further investigation. Perhaps obesity-related factors influence the immune response differently in this context. The authors shall provide possible explanations for this observation.

-Although the study evaluated anti-Nucleoprotein antibodies for identification of reinfection and breakthrough infection in Pacific Island cohorts, it did not directly compare titer levels across different ethnic groups. It would indeed be interesting to explore these differences further. In addition, the kinetics of anti-Nucleoprotein antibodies over time were not explicitly discussed in the study. Investigating the temporal changes in these antibodies could provide valuable insights.

Reviewer #4: See attachment

Michael Dewey

[LINK]

COMMENTS FROM THE ACADEMIC EDITOR:

The Academic Editor thought very favourably of your study from the outset, stating that he believed that your paper was putting forward priority work. He commented that you ‘generated meaningful data that has relevance to public health policy’. He has since read the reviewer comments and maintains that your paper addresses a definite gap in the published literature. He was in favour of sending this back to you to proceed to major revision.

GENERAL EDITORIAL REQUESTS:

1. Please upload any figures associated with your paper as individual TIF or EPS files with 300dpi resolution at resubmission; please read our figure guidelines for more information on our requirements: http://journals.plos.org/plosmedicine/s/figures. While revising your submission, please upload your figure files to the PACE digital diagnostic tool, https://pacev2.apexcovantage.com/. PACE helps ensure that figures meet PLOS requirements. To use PACE, you must first register as a user. Then, login and navigate to the UPLOAD tab, where you will find detailed instructions on how to use the tool. If you encounter any issues or have any questions when using PACE, please email us at PLOSMedicine@plos.org.

To submit your revised manuscript please use the following link:

---

## [Decision Letter · Decision Letter 2]

10 Jul 2024

Dear Dr Inizan,

Many thanks for submitting your manuscript "Levels and functionality of Pacific Islanders’ hybrid humoral immune response to BNT162b2 vaccination and Delta/Omicron infection: a cohort study in New Caledonia" (PMEDICINE-D-24-01093R2) to PLOS Medicine. The paper has been reviewed by subject experts and a statistician; their comments are included below and can also be accessed here: [LINK]

As you will see, there were some significant concerns highlighted by the statistical reviewer (Reviewer 4). After discussing the paper with the editorial team and an academic editor with relevant expertise, I'm pleased to invite you to revise the paper in response to the reviewers' comments. We plan to send the revised paper to some or all of the original reviewers, and we cannot provide any guarantees at this stage regarding publication.

We ask that you submit your revision by Jul 31 2024 11:59PM. However, if this deadline is not feasible, please contact me by email, and we can discuss a suitable alternative.

Don't hesitate to contact me directly with any questions (ssunny@plos.org). 

Best regards, 

Syba 

Syba Sunny, MBBS, MRes, FRCPath 

Associate Editor

PLOS Medicine

ssunny@plos.org

Comments from the editorial team:

Thank you again for submitting this revised manuscript. We continue to find your study interesting, but are concerned about the level of missingness of key information on how the study was conducted. We are in agreement with the points raised by the reviewers and, especially, the statistician (Reviewer 4), and ask that you address all points in full. Given the significance of the aforementioned issues, we note that not addressing these concerns sufficiently would likely result in the journal choosing not to take your manuscript any further.

In addition to the above, we have listed some smaller points for you to address below:

(1) It is our understanding that you detected anti-spike antibodies in 100% of participants – please report this in your abstract.

(2) We see no need to abbreviate New Caledonia to NC; we would prefer if you would be so kind to spell it out throughout. 

(3) We ask that ‘nucleoprotein,’ ‘spike,’ ‘glycoprotein’ (etc) not be capitalized, nor SARS-CoV-2 variants (e.g. alpha, delta, omicron - Wuhan is an exception). 

(4) Line 125 - please remove the word ‘deployed’ and revise the sentence so that it reads ‘Humoral and cellular responses among Australian First Nations people were similar…’ or something similar.

(5) Lines 138-140: ‘We aimed to characterize Pacific Islanders’ hybrid humoral response to BNT162b2 vaccine and SARS-CoV-2 infection, by evaluating anti-Spike (anti-S) IgG levels, the neutralization capacity…’ – please also mention anti-nucleocapsid antibodies in this sentence.

Comments from the academic editor:

The Academic Editor agreed with the comments raised by the statistical reviewer and believed these were important to address. He supported the decision to send your manuscript back to you for a further revision.

Comments from the reviewers: 

Reviewer #1: I thank the authors for attending to my previous comments so comprehensively. There are just a couple of comments.

1. Whilst the authors have included a sentence in the methods section line 164166 page 8 clarifying the inclusion criteria for timings between doses. It would be useful to include the mean and range between each dose, as this is important information

needed when interpreting the data.

2. The paper by Hartley et al. shows SARS-CoV-2-specific IgG levels in individuals plasma. This is quantified to different SARS-CoV-2 strains hence the query to whether this had been quantified e.g. in ELISA. 

3. For neutralisation, please specify the change in neutralisation levels between doses, so that it is clear whether this changes between doses. Also, specify % patients with positive Spike-specific IgG to show whether

seroconversion levels change between doses.

Reviewer #3: I have no further comments/remarks. All the comments and remarks I made in my previous review were appropriately addressed.

Reviewer #4: See attachment

---

---

## [Decision Letter · Decision Letter 3]

23 Aug 2024

Dear Dr. Inizan,

Thank you very much for re-submitting your manuscript "Levels and functionality of Pacific Islanders’ hybrid humoral immune response to BNT162b2 vaccination and delta/omicron infection: a cohort study in New Caledonia" (PMEDICINE-D-24-01093R3) for review by PLOS Medicine.

I have discussed the paper with my colleagues and the academic editor and it was also seen again by 1 subject reviewer and 1 statistician. Reviewer 1 (the subject reviewer) brought up some points that will need addressing (see both the academic editor’s comments as well as Reviewer 1’s report to understand which points exactly need addressing). However, other than this, and provided the remaining editorial and production issues are dealt with, I am thrilled to inform you that we are planning to accept the paper for publication in the journal.

In revising the manuscript for further consideration here, please ensure you address the specific points made by the reviewer and the editors. In your rebuttal letter you should indicate your response to the reviewer’s and editors' comments and the changes you have made in the manuscript. Please submit a clean version of the paper as the main article file. A version with changes marked must also be uploaded as a marked up manuscript file.

We expect to receive your revised manuscript within approximately 1 week or so. Please email us (plosmedicine@plos.org) if you have any questions or concerns.

We look forward to receiving the revised manuscript by Sep 04 2024 11:59PM. 

Sincerely,

Syba

Syba Sunny, MBBS, MRes, FRCPath

Associate Editor 

PLOS Medicine

ssunny@plos.org

EDITORIAL COMMENTS:

Thank you again for engaging so thoroughly with the reviewer and editorial comments. We appreciate the value of the data presented here and are confident that this will make an impact on an underrepresented population.

As noted earlier in this email, Reviewer 1 made a number of points. We consulted the academic editor on this review and he asked that you kindly address 2 of the reviewer’s comments (see below). He believed that the changes required would be unlikely to make a change to the overall conclusions and, hence, supported a minor revision outcome at this stage. 

The academic editor recommended that the authors address the following:

(1) ‘It has been shown in numerous publications that neutralisation of Omicron BA.1 and BA.4/5 differ both at post dose 2 and post dose 3 of COVID-19 vaccination. In lines 350-355 the authors provide a blanket statement that neutralisation of omicron increases from 19.1% at dose 2 to 97.9 at dose 3. Which omicron variant is this to and are there differences in neutralisation of each variant as has been previously published.’ He asked that some comment/clarification be provided here, with no need for further experiments, etc.

(2) ‘Figure 3C is not easy to understand and why does the scale for % neutralisation only go up to 1 when you report in the text that the neutralisation went up to 97.9 at dose 3. I would suggest graphing this in a different way to make it easier to understand.’ 

REVIEWER COMMENTS:

Reviewer #1: Whilst the authors have attended to most of the comments I posed, there are a still a number of outstanding issues that require attention prior to publication.

- It has been shown in numerous publications that neutralisation of Omicron BA.1 and BA.4/5 differ both at post dose 2 and post dose 3 of COVID-19 vaccination. In lines 350-355 the authors provide a blanket

statement that neutralisation of omicron increases from 19.1% at dose 2 to 97.9 at dose 3. Which omicron variant is this too and are there differences in neutralisation of each variant as has been previously 

published.

-Is there any information of infection status of the subjects at each dose as this can have an impact on the level of anti-spike IgG and neutralisation titers measured in the serum. If not this should be stated. Did infection have an impact

on the magnitude of response observed.

- Figure 3C is not easy to understand and why does the scale for % neutralisation only go up to 1 when you report in the text that the neutralisation went up to 97.9 at dose 3. I would suggest graphing this in a different way to make it easier to understand.

Reviewer #4: The authors' rebuttal has addressed my concerns.

EDITORIAL REQUESTS:

Abstract: Please combine the Methods and Findings section into one section with the heading ‘Methods and Findings’.

Data Availability Statement: Are you able to provide an email address that is not for the corresponding author but perhaps to a generic inbox? Directing data requests to a non-author institutional point of contact, such as a data access or ethics committee, helps guarantee long term stability and availability of data. Providing interested researchers with a durable point of contact ensures data will be accessible even if an author changes email addresses, institutions, or becomes unavailable to answer requests.

Author Summary: 

(1) Could you expand the first sentence for our readers please? ‘Pacific Islanders are at risk for severe COVID-19’ – could you add a note here to explain why this is the case for the non-specialist reader?

(2) In the sentence ‘Characterizing the levels and functionality of Pacific Islanders humoral immune response to Pfizer BNT162b2 vaccine is pivotal to adapt vaccination regimen to regional singularities’ – I suggest changing ‘regimen’ to ‘regimens’ and adding a full stop at the end of the sentence.

(3) In the sentence ‘The current study enrolled 585 participants from New Caledonia self-declaring as Melanesians, Polynesians, Europeans or belonging to Other Communities and characterized their antibodies levels and functionality at one and three months post second dose or one and six months post-third dose of the Pfizer BNT162b2 vaccine’, it may be best to break this up a little to improve readability. Perhaps something like this (or similar): ‘The current study enrolled 585 participants from New Caledonia self-declaring as Melanesians, Polynesians, Europeans or belonging to Other Communities. Antibody levels and their functionality were characterized at the following time-points: (1) one and/or three months after a second dose of the Pfizer BNT162b2 vaccine was administered, or (2) one and/or six months after the third dose of the vaccine.’

(4) Perhaps include the word ‘all’ before ‘sera’ in the sentence ‘61.3% of the sera contained traces of previous SARS-CoV-2 infection; the cohort’s immune response was thus mostly hybrid, resulting from both vaccination and infection’.

(5) As the Author Summary is meant to be non-technical and accessible to a non-expert audience, please revise the following sentence by explaining (briefly) what the relevance of ‘anti-spike IgG levels’ are and also explain the abbreviation ‘ADCC’. The sentence in question is: ‘At all timepoints analyzed, the hybrid humoral immune response remained equivalent across the Melanesian, Polynesian, European and Other communities: anti-spike IgG levels, capacity to mediate omicron neutralization and capacity to mediate ADCC were not significantly different across communities (p values>0.05).’

Introduction:

Line 109: ‘Considering ethnic representation in COVID-19 vaccine trials is crucial.’ It might be useful to expand on this sentence so it matches what you eventually write in the Author Summary.

Discussion:

Line 493-495: ‘Second, the high rate of SARS-CoV-2 infections prevented to assess ethnicity’s impact…’ – would this be better written as ‘…prevented us from being able to assess…’ or something of that ilk? Forgive me if I’ve misinterpreted the meaning behind this sentence.

Line 522: Please replace the semi-colon with a comma.

---

## [Editor Report · Decision Letter 4]

30 Aug 2024

Dear Dr Inizan, 

On behalf of my colleagues, I am thrilled to inform you that we have agreed to publish your manuscript "Levels and functionality of Pacific Islanders’ hybrid humoral immune response to BNT162b2 vaccination and delta/omicron infection: a cohort study in New Caledonia" (PMEDICINE-D-24-01093R4) in PLOS Medicine.

PRESS

Sincerely, 

Syba

Syba Sunny, MBBS, MRes, FRCPath 

Associate Editor 

PLOS Medicine

ssunny@plos.org